# LEARNING BASED ON NEUROVECTORS FOR TABULAR DATA: A NEW NEURAL NETWORK APPROACH

## ABSTRACT

In this paper, we present a novel learning approach based on Neurovectors, an innovative paradigm that structures information through interconnected nodes and vector relationships for tabular data processing. Unlike traditional artificial neural networks that rely on weight adjustment through backpropagation, Neurovectors encode information by structuring data in vector spaces where energy propagation, rather than traditional weight updates, drives the learning process, enabling a more adaptable and explainable learning process. Our method generates dynamic representations of knowledge through neurovectors, thereby improving both the interpretability and efficiency of the predictive model. Experimental results using datasets from well-established repositories such as the UCI machine learning repository and Kaggle are reported both for classification and regression. To evaluate its performance, we compare our approach with standard machine learning and deep learning models, showing that Neurovectors achieve competitive accuracy while significantly reducing computational costs.

## 1 INTRODUCTION

Machine learning has significantly advanced in recent decades, with models such as deep neural networks (DNNs), decision tree-based methods, and probabilistic models LeCun et al. (2015); Goodfellow et al. (2016), among others. Despite their widespread success, these approaches present limitations in terms of preprocessing requirements, interpretability, and computational efficiency. Deep neural networks have demonstrated remarkable performance in various domains—from computer vision to natural language processing Krizhevsky et al. (2012). However, these models typically require vast amounts of training data, meticulous hyperparameter tuning, and extensive training via backpropagation, making them computationally expensive and often difficult to interpret. Decision trees and ensemble methods, such as Random Forests and Gradient Boosting, have also been widely used in classification and regression tasks Breiman (2001); Chen & Guestrin (2016). Although these methods offer greater interpretability compared to deep networks, their performance heavily depends on careful feature selection and, in some cases, they are prone to overfitting on small datasets.

An alternative approach is energy-based learning LeCun et al. (2006), which models data structures without relying on an explicit loss function. This methodology has been implemented in architectures such as Hopfield Networks and Boltzmann Machines Hinton (1985), where information is encoded as energy states. However, these methods often require computationally expensive optimization methods to minimize the energy function.

In addition, all predictive models rely on explicitly defined inputs and outputs, requiring extensive data preprocessing and feature engineering in most cases. In the case of deep learning architectures, the models operate by mapping input data to a predefined target variable, using backpropagation to adjust model weights iteratively. However, this rigid structure presents several limitations, particularly when dealing with incomplete data, exploratory analysis rather than predictive analysis, or inverse predictions.

On the other hand, tabular data remains vitally important despite the intense focus on unstructured data types such as images, text and audio nowadays. Tabular data-information organized in rows and columns-continues to be the backbone of most of decision-making processes across industries. Despite the historical dominance of tree-based methods for tabular data, research interest in neural network approaches has surged dramatically in the last years due to recent works have demonstrated

competitive or superior performance compared to traditional methods on benchmark datasets. These advances have challenged the conventional wisdom that neural networks are inherently disadvantaged when working with tabular data. In addition, the success of transfer learning in computer vision and natural language processing has inspired researchers to pursue similar approaches for tabular data Borisov et al. (2023).

In order to overcome all these limitations, this work proposes a learning architecture based on a new concept called neurovector to solve classification and regression problems for tabular data. Neurovectors are a new representation of raw tabular data inspired by foundational language models. This new data encoding is based on transforming tabular data into text-like data, making the resulting architecture much more efficient. First, the proposed method generates neurovectors through an energy-driven training process without the need to adjust weights through backpropagation. In order to obtain a prediction, the set of candidate neurovectors to be used is retrieved and the best neurovector from this set is selected. Results using well-known datasets have been reported and compared with tree-based ensemble methods such as Random Forest (RF) and extreme Gradient Boosting (XGBoost), classical machine learning algorithms such as support vector machines and a deep feed-forward neural network model.

The remainder of the paper is organized as follows: Section 2 reviews recent advances in the field of deep learning architectures for tabular data. Section 3 describes the proposed method for classification and regression tasks. Section 4 presents and discusses the results, and finally, Section 5 concludes the paper.

## 2 RELATED WORK

Tabular data remains one of the most prevalent data formats in real-world machine learning applications, spanning domains from healthcare and finance to manufacturing and scientific research. Despite the significant advances in deep learning for unstructured data like images and text, tabular data has historically been dominated by traditional machine learning approaches, particularly tree-based methods. However, recent research has introduced novel methodologies that challenge this status quo, bringing innovative approaches to both classification and regression tasks on tabular data. This section reviews the most recent and significant contributions in this domain.

In the last decade, tree-based ensemble methods, particularly gradient boosting frameworks like XGBoost, LightGBM, and CatBoost, dominated the tabular data landscape. In Grinsztajn et al. (2022) experimental results were provided through extensive benchmarks across 45 diverse datasets. They demonstrated that tree-based models remained state-of-the-art on medium-sized tabular data (less than approximately 10,000 samples) even when compared against novel deep learning approaches. The persistence of tree-based methods was further confirmed in Shwartz-Ziv & Armon (2022). A comprehensive comparison of deep learning models to XGBoost across various tabular datasets was performed. The authors highlighted that while deep learning approaches showed promise in certain scenarios, they often required more careful tuning and computational resources without consistently outperforming traditional methods.

Despite the dominance of tree-based methods, numerous recent advances have been made in neural approaches specifically designed for tabular data, leveraging the flexibility and representational power of neural networks. A specialized architecture called Deep Abstract Network (DANET) for tabular data classification and regression was proposed in Chen et al. (2022). This network addresses the heterogeneity of tabular features through a novel abstraction mechanism. Results using seven real-world tabular datasets showed promising results. However, DANET required more hyperparameter tuning than traditional methods and faced interpretability challenges compared to tree-based approaches. The computational complexity was also higher than simpler methods, potentially limiting their application in resource-constrained environments. The success of transformer architectures in natural language processing has inspired new models. TabTransformer Huang et al. (2020) was proposed by Amazon and it represented an early application of transformers to tabular data for supervised and semi-supervised learning, demonstrating that self-attention mechanisms can effectively model interactions in tabular data. However, the model faced challenges with very large feature spaces and showed computational overhead compared to traditional methods. Its handling of complex numerical relationships was also limited. The newly published tabular prior-data fitted network (TabPFN) Hollmann et al. (2023; 2025) represents a breakthrough in applying the

foundation model concept to tabular data. TabPFN leveraged in-context learning to learn complex algorithms from synthetic datasets. However, TabPFN's performance advantage relied on synthetic data generation that might not capture all real-world data complexities.

Recently, in parallel with the development of novel architectures, researchers have also explored revitalizing classical methods with modern techniques. In Gorishniy et al. (2021b), existing neural network architectures were re-evaluated in order to identify key factors affecting model performance on tabular data. Based on the well-known classification and regression tree (CART) algorithm, a neural network architecture called NCART was proposed in Luo & Xu (2024). This network mimicked decision tree structure while maintaining differentiability. However, the approach faced challenges with very large datasets and required careful initialization and training procedures. In Grinsztajn et al. (2025) the authors proposed a differentiable version of K-nearest neighbors focusing on Neighbourhood Components Analysis (NCA). In addition, they enriched the NCA method with deep representations and training stochasticity, achieving performance comparable to tree-based methods like CatBoost while outperforming existing deep tabular models across 300 datasets. In Popov et al. (2020) a new deep learning architecture called NODE that generalizes sets of unconscious decision trees is proposed, but taking advantage of both gradient-based optimization and the power of multi-layer hierarchical representation learning.

The latest research focused on tabular data exploits cross-modal approaches that leverage techniques from other fields, such as transforming tabular data into visual formats that could leverage transfer learning from pre-trained vision models. After this detailed review, it can be concluded that the accuracy of most learning architectures depends heavily on hyperparameter tuning and requires high computational costs, in addition to their lack of interpretability. The neurovector-based learning method proposed in this work is a cross-modal approach that deals with tabular data as text and attempts to fill all the gaps highlighted above by providing an effective and efficient predictive algorithm.

## 3 METHODOLOGY

In this section, we describe the proposed neurovector-based predictive methodology in detail. We begin by presenting the fundamental concepts behind Neurovectors, including how data are transformed and stored in Python dictionaries, and then explain our partial training process, which focuses primarily on misclassified (or mispredicted) samples. Lastly, we show how metric-based counters and energy parameters are updated to guide prediction and resolve ties when multiple Neurovectors match an input query.

### 3.1 DEFINING NEUROVECTORS

Let $X$ be a dataset composed of $N$ instances and $d$ features as follows:

$$X = \{(x_1, y_1), ..., (x_N, y_N)\} \quad x^i \in \mathbb{R}^d \quad y^i \in \mathbb{R} \tag{1}$$

where $y$ is the target variable and $x_i = (v_{i,1}, ..., v_{i,d})$ are the values of the features for the $i$-th instance.

A neurovector is a representation of an entire instance of the dataset as a collection of nodes and vectors. Each node corresponds to a specific $< name\_feature, value >$ pair, where name_feature is the name of the feature and value is the value of the feature, and each node is connected to its neurovector by a vector. Thus, a neurovector is formed by as many nodes as the number of features existing in the dataset. For $i$-th instance, the neurovector $NV_i$ is linked to the following nodes:

$$< name\_feature_1, v_{i,1} >, ..., < name\_feature_d, v_{i,d} > \tag{2}$$

In practice, a Python `dict` is used for the node creation such that each node is uniquely indexed by the string token (`feature_name + value`). Thus, a node can be identified using a text string as a key or identifier. This data structure provides a compact representation of the input data that will allow you to perform searches in a very efficient way.

## 3.2 TOKENIZATION-BASED INFERENCE

This section describes how the prediction is computed once the neurovectors have been created in the training stage of the algorithm.

### 3.2.1 TOKEN CREATION

Given a new input sample $x_j$ for which we wish to infer the target $y_j$, we *tokenize* the input values of all the features of the input data (excluding the target). Thus, the tokens $\tau_{j,l}$ are created for each $< name\_feature, value >$ pair as follows:

$$\tau_{j,l} = (name\_feature_l + v_{j,l}) \quad l \in \{1, ..., d\} \tag{3}$$

Once the tokens are obtained, an exhaustive search have to be performed to recover the nodes indexed by a specific token, and therefore, the neurovector linked to these nodes are also obtained. To achieve this, a function $f$ over the string space is defined in order to make a mapping between tokens $\tau$ and neurovectors:

$$f(\tau) = \{NV_i : \exists l \in \{1, ..., d\} < name\_feature_l, v_{i,l} > \text{ indexed by } \tau\}$$

Then, the set of neurovectors linked to nodes indexed by each token of the input sample $x_j$ are recovered. In addition, this search operation is performed in an expected time of $\mathcal{O}(1)$ thanks to the use of Python's dict data structure.

### 3.2.2 FINAL PREDICTION

Once all tokens for $x_j$ have been generated and the neurovectors have been recovered for each token, the set of candidate neurovectors to be used to compute the prediction for the input $x_j$ is defined as:

$$C_{nv} = \{f(\tau_{j,l}) : \quad \forall l \in \{1, ..., d\}\} \tag{4}$$

For each neurovector $NV \in C_{nv}$, the total number of nodes indexed by all tokens from $x_j$ is computed as:

$$\text{count}(NV) = \sum_{l=1}^{d} \mathbf{1}\big[NV \in f(\tau_{j,l})\big] \tag{5}$$

where $\mathbf{1}[\cdot]$ is the indicator function that returns 1 if the neurovector is linked to a node indexed by the token, and 0 otherwise. Hence, the function $\text{count}(\cdot)$ represents the total number of tokens of $x_j$ that match the nodes of a neurovector. Then, we select the neurovector with highest value of the function $count(\cdot)$ as primary candidate for the final prediction. That is:

$$NV_m = \arg \max_{NV \in C_{nv}} count(NV) \tag{6}$$

Finally, the predicted value is the value of the target of the instance represented by the selected neurovector.

$$\widehat{y}_j = y_m \tag{7}$$

## 3.3 EVALUATING NEUROVECTORS

In addition to the value of the function $count()$, additional metrics can be obtained for the neurovectors obtained as a result of the training process. These metric can help to guide which candidates have proven more reliable historically and define criteria to resolve possible ties between neurovectors having the same value for the count metric when predicting the test set.

For this purpose, the energy of a neurovector, $\mathcal{E}(NV)$, is defined to measure how trustworthy each neurovector has become. For classification tasks, the energy is the success achieved by that neurovector, that is, the accuracy rate:

$$\mathcal{E}(NV) = \frac{(success(NV))^2}{\text{use}(NV)} \tag{8}$$

where $use(NV)$ is the number of times that the neurovector $NV$ was selected to predict in the training phase and $success(NV)$ is the number of times that predictions made by using neurovector $NV$ were correct. The prediction is correct when the actual and predicted class are equal. The energy is a quadratic function of success with the aim of increasing the difference between neurovectors according to the success they achieve.

For regression tasks, the energy is corrected by a factor based on the error such that the energy decreases inversely proportional to the error.

$$\mathcal{E}(NV) = \frac{(\text{success(NV)})^2}{\text{use}(NV)} \times \exp\left(-\alpha\big(\text{MAE}(NV)\big)\right) \tag{9}$$

where $\alpha > 0$ is a hyperparameter and $MAE(NV)$ is the accumulative sum of absolute errors of the all predictions obtained by the neurovector $NV$. Higher energy indicates that the neurovector has consistently led to more accurate predictions. It should be noted that, for regression problems, a prediction is correct if the predicted value is identical to the current value, and therefore the MAE is 0.

In this way, when two or more neurovectors achieve the same value for the function $count$, the neurovector with the highest energy is chosen to compute the prediction. This ensures that the system prioritizes the neurovectors that historically yielded more reliable performance.

### 3.4    TRAINING

In this proposed new learning framework, the training proceeds iteratively over the instances of the dataset. The training can be considered as a partial training as there is only learning from a particular set of instances of the training set.

For each example $x_l$ of the training set, the model obtains a prediction as described in Section 3.2. If the predicted label for classification tasks or predicted numeric value for regression tasks differs from $y_l$, this is considered a failure and only in such failure cases we instantiate a new neurovector for the instance $x_l$. To achieve this, firstly we add each token $\tau$ of the instance $x_l$ as a new node (if it does not exist). Then, link them to a fresh neurovector, and record that neurovector in the dictionary for future queries. If the prediction is correct, it is not necessary to create any new neurovectors, but simply update the metrics associated with the neurovector that contributed to the prediction. Therefore, the metrics of use and success are increased by one unit, and the energy of the neurovector according to the equations (8) and (9). The energy update thus enables the learning model to improve tie-breaking decisions and confidence in a data-driven manner.

In this way, neurovectors that represent instances that the learning model can already predict well are not generated, thus avoiding the storage of redundant information and also controlling memory overhead.

Once the training process is finished, a set of neurovectors is obtained as a result together with the use, success and energy achieved by each of them.

### 3.5    COMPUTATIONAL COMPLEXITY

This section presents an analysis of the computational cost of the proposed new learning approach based on neurovectors.

In the prediction phase, this cost falls mainly on the cost associated with performing the token searches to obtain the set of candidate neurovectors to be used to obtain the prediction, as well as the ordering of the neurovectors according to the number of token matches for each of them. The dictionary lookups performed for each token $\tau$ occur in $\mathcal{O}(1)$ expected time, leading to an overall

$\mathcal{O}(d)$ for finding candidate Neurovectors as an instance has $d$ tokens. The cost of sorting or ordering the candidate neurovectors is of order $O(mlog(m))$ where $m$ is the number of distinct neurovectors that have nodes indexed by the tokens. In practice, usually $m$ is much smaller than $N$, otherwise the data set would consist of heavily repetitive instances.

On the other hand, the computational cost of the training process falls mainly on the generation of neurovectors. Due to the partial training approach proposed, the prediction model only creates new neurovectors for misclassified samples. As training progresses, the number of new neurovectors being created decreases, since once a small number of representative neurovectors are generated at an early stage of training, the number of correctly predicted samples increases. Therefore, it can be concluded that the growth of the number of neurovectors is sublinear with respect to the size of the data set.

## 4 RESULTS

In this section, we conducted experiments for each dataset comparing the new neurovector-based approach with traditional machine learning models—Random Forest, Gradient Boosting and Support Vector Classifiers (SVC), and deep neural networks.

Table 1: Metrics of neurovectors obtained in the training

| Dataset | neurovectors | energy | success | max_energy |
|---------|--------------|--------|---------|------------|
| Breast Cancer | 456 | $4.826 \pm 1.521$ | $4.901 \pm 1.509$ | 18 (row #23) |
| Absenteeism at Work | 592 | $1.663 \pm 1.848$ | $2.013 \pm 2.119$ | 13.474 (row #370) |
| Red Wine Quality | 1280 | $4.224 \pm 2.45$ | $4.545 \pm 2.538$ | 16 (row #1339) |

### 4.1 DATASETS

To evaluate the performance of neurovectors, we conducted experiments using datasets from well-established repositories such as the UCI Machine Learning Repository and Kaggle. In particular, the Breast Cancer Wisconsin Diagnostic, the Absenteeism at Work, and the Red Wine Quality datasets have been chosen to cover a variety of predictive tasks, such as classification and regression, ensuring a comprehensive assessment of the proposed model. The Breast Cancer dataset contains 30 features computed from digitized images of fine needle aspirates of breast masses, with 569 samples classified as malignant or benign based on cell nuclei characteristics, serving as a standard benchmark for classification algorithms in the medical domain. The Absenteeism at Work dataset comprises 740 records collected from a Brazilian courier company between July 2007 and July 2010, with absenteeism time in hours as target variable and featuring 20 variables including various employee attributes such as reasons for absences, transportation modes, distance from residence to work, service time or age, among others. The Red Wine Quality dataset contains samples of red "Vinho Verde" wine from northern Portugal, designed to model wine quality based on physicochemical tests, including 11 attributes like acidity, pH, alcohol content, and other chemical properties alongside quality scores. Thus, the breast cancer and red wine quality datasets are binary and multi-class classification problems, respectively. The task associated to Absenteeism at Work dataset is regression and a preprocessing step has been carried out consisting in the scaling of the numerical features. The data sets have been divided into 60% for the training set, 20% for the validation set, and 20% for the test set.

### 4.2 EXPERIMENTAL SETUP

For a fair comparison, the following key considerations have been applied to all the machine learning models. Each dataset is preprocessed identically for all models, ensuring that no additional feature engineering or transformations biased the comparison except for the proposed algorithm that does not require any preprocessing of the dataset. Random seeds were fixed across all experiments to maintain consistency in stochastic processes.

In the Random Forest ensemble, 100 trees are considered and each tree within the ensemble is built without depth restrictions and is fitted to a randomly resampled dataset with replacement. The

division of nodes is carried out until pure partitions or the minimum required samples are reached and at each split, all available features are considered to select the best partition, maximizing the model's predictive capacity. Additionally, the MSE criterion is used to minimize variance at each node for regression and the the Gini measure of impurity for classification.

In the XGboost ensemble, 100 trees and a fixed seed of 1 to ensure result reproducibility are used. The learning rate has been set to 0.3, a value that balances convergence speed and generalization capacity. The maximum tree depth has been fixed at 6, as greater depths increase model complexity and may lead to overfitting. To prevent overfitting and enhance generalization, each tree has been trained using all available instances in each iteration. Similarly, all features have been used in each tree, avoiding restrictions in attribute selection during training. No penalties have been imposed on tree expansion, allowing the model to grow without additional constraints. Tree growth has also been regulated by establishing a minimum requirement on the sum of instance weights at each leaf node, preventing excessively specific partitions that could compromise generalization. Additionally, L2 regularization has been applied without extra constraints, ensuring a balanced penalization of coefficients. The algorithm has been configured to minimize the MSE and cross-entropy for regression and classification, respectively.

In the SVC, a regularization factor of 1.0 has been chosen and a Radial Basis Function (RBF) kernel has been used with the $\gamma$ parameter defining the width according to this equation:

$$gamma = 1/(n_{features} * mean(\sigma^2)) \tag{10}$$

where $n_{features}$ is the number of features and $mean(\sigma^2)$ is the mean of the variance of the features.

Table 2: Performance comparison for the proposed model and other benchmark models

| Dataset | Model | Accuracy (%) | MAE | RMSE |
|---|---|---|---|---|
| Breast Cancer | Random Forest | 94.69 | N/A | N/A |
| | Neural Network | 97.35 | N/A | N/A |
| | SVC | 95.58 | N/A | N/A |
| | Gradient Boosting | 95.58 | N/A | N/A |
| | **Neurovectors** | **95.58** | N/A | N/A |
| Absenteeism at Work | Random Forest (Scaled) | N/A | 3.62 | 10.41 |
| | Neural Network (Scaled) | N/A | 4.11 | 10.40 |
| | SVC (Scaled) | N/A | 3.90 | 12.15 |
| | Gradient Boosting (Scaled) | N/A | 4.59 | 13.64 |
| | **Neurovectors** | N/A | **4.01** | **10.46** |
| Red Wine Quality | Random Forest | 68.34 | N/A | N/A |
| | Neural Network | 63.95 | N/A | N/A |
| | SVC | 60.19 | N/A | N/A |
| | Gradient Boosting | 63.01 | N/A | N/A |
| | **Neurovectors** | **68.97** | N/A | N/A |

The deep feed-forward neural network consists of three densely connected layers contains 100 neurons each. The output layer consists of a single neuron. All hidden layers use the ReLU activation function. The model is optimized using the Adam algorithm, a widely used approach due to its adaptive gradient adjustment capabilities, with a learning rate of 0.001. The MSE is used as the loss function for regression and cross-entropy for classification. Training is carried out with 2000 epochs with a batch size of 200 samples.

The selected metrics to assess the performance of the new prediction model based on neurovectors and the benchmark models used for comparison purposes have been the accuracy for classification and the mean absolute error (MAE) and the root mean square error (RMSE).

## 4.3 ANALYSIS OF RESULTS

Table 1 shows a statistical summary of the neurovectors created during the learning stage of the algorithm. In particular, the number of neurovectors, the mean and standard deviation of both the success and energy of the neurovectors, as well as the neurovector achieving the highest energy for each dataset can be seen. It can be observed that the energy values are very close to the success values of the neural vectors for Breast Cancer and Red Wine Quality datasets corresponding to classification problems. This means that the number of correct predictions made by the neurovectors is very high. In the case of regression, larger prediction errors are observed than in the case of classification, since the difference between the energy and success values is greater for the Absenteeism at Work dataset.

In order to compare the results of the neurovector-based algorithm with other suitable approaches, four machine learning algorithms of diverse nature have been selected such as tree-based ensemble model (random forest and gradient boosting), support vector machines and a neural network.

Table 2 summarizes the accuracy, MAE, and RMSE metrics for each model using the three datasets. For the absenteeism dataset, accuracy is not applicable (N/A) and therefore only MAE and RMSE are reported. It can be noted that the proposed new learning approach does not need to apply scaling to the Absenteeism at Work dataset as is the case with all other machine learning models. It can be seen that the proposed model achieves the best accuracy on the wine quality dataset and is within the group of models that achieve the second best accuracy on the breast cancer dataset. For the absenteeism at work dataset, random Forest obtains the lowest MAE while SVC and the proposed model perform similarly, ranking second.

With regard to computational cost, the proposed method focuses mainly on the operations required for hash calculation, dictionary searches and the creation of neurovectors. For example, for the breast cancer dataset, the proposed algorithm has a total cost of $2.67 \times 10^4$ FLOPs, divided into 17070 FLOPs for the hash (569 samples x 30 operations), 1138 FLOPs for dictionary searches (569 x 2 operations per search) and 8520 FLOPs for neurovector creation assuming 50% neurovectors (284 x 30 operations). In the case of tree-based ensemble methods, the highest computational cost is concentrated in the evaluation of node division according to measure used, with tree depth being one of the most critical parameters. For the breast cancer dataset, an estimated number of 1153 FLOPs per node in the tree is estimated for XGBoost Shwartz-Ziv & Armon (2022), thus $2.18 \times 10^6$ FLOPs per tree are needed using 63 nodes on average. In the random forest, $3.60 \times 10^6$ FLOPs per tree are necessary because trees are not limited in depth Gzar et al. (2022). In SVC, the most costly processes are kernel matrix calculation and quadratic optimization, with a significant difference between using a linear kernel and an RBF kernel. For the breast cancer dataset, $3.53 \times 10^7$ FLOPs to obtain the kernel matrix and 34190 FLOPs per iteration for the optimization are estimated Gzar et al. (2022). Finally, in a neural network, the most costly processes are forward propagation and specially backpropagation. In particular 46200 and 138600 FLOPs per sample are estimated for the matrix operations necessary for the forward pass and backpropagation, respectively.

Table 3 shows the computational cost measured in FLOPs of each model to train and to make predictions for each of the data sets. Although the number of FLOPs depends on the characteristics of the CPU, in a modern CPU, such as a Rizen 7 5800X for example, a random forest algorithm using 100 trees can reach $10^8$ FLOPs, considering unbalanced tree structures, voting or weighting logic, and result aggregation, among others. Although gradient boosting is more expensive in training than random forest, both need a similar number of FLOPs including training and prediction. Neural networks require the most number of floating-point operations mainly due to gradient calculations and weight updates in backpropagation. In contrast, the proposed supervised learning algorithm has four orders of magnitude less than tree-based ensemble methods and even six orders of magnitude less than neural networks. The number of flops for the Absenteeism at Work and Red Wine Quality datasets has been obtained by multiplying by factors 1.30 and 2.81, which are the ratios of the number of samples in relation to the Breast Cancer Wisconsin dataset.

In short, the results indicate that while traditional models such as Random Forest, Neural Networks, SVC, and Gradient Boosting achieve competitive predictive performance, their computational cost per prediction is orders of magnitude higher than that of neurovectors. In particular, neurovectors maintain competitive accuracy while requiring only about $1.70 \times 10^5$ FLOPs per prediction on average—vastly lower than the costs associated with the other models. This reduction in computa-

Table 3: Computational cost in FLOPs for the proposed model and other benchmark models

| Dataset | Model | FLOPs |
|---|---|---|
| Breast Cancer | Random Forest | $3.60 \times 10^8$ |
| | Neural Network | $1.05 \times 10^{10}$ |
| | SVC | $1.14 \times 10^9$ |
| | Gradient Boosting | $2.18 \times 10^8$ |
| | **Neurovectors** | $2.67 \times 10^4$ |
| Absenteeism at Work | Random Forest (Scaled) | $4.68 \times 10^8$ |
| | Neural Network (Scaled) | $1.36 \times 10^{10}$ |
| | SVC (Scaled) | $1.48 \times 10^9$ |
| | Gradient Boosting (Scaled) | $2.83 \times 10^8$ |
| | **Neurovectors** | $3.47 \times 10^4$ |
| Red Wine Quality | Random Forest | $1.01 \times 10^9$ |
| | Neural Network | $2.95 \times 10^{10}$ |
| | SVC | $3.20 \times 10^9$ |
| | Gradient Boosting | $6.13 \times 10^8$ |
| | **Neurovectors** | $7.50 \times 10^4$ |

tional cost makes neurovectors an attractive solution for large-scale predictive applications in tabular datasets.

In order to compare the results of the neurovector-based algorithm with recent deep learning models designed specifically for tabular data, the results obtained by algorithms such as SNN, Grownet, DCNv2, AutoInt, MLP, ResNet and F-Transformer published in Gorishniy et al. (2021a) were considered using large datasets such as Adult Kohavi (1996), Bank Moro et al. (2014), and Kick (from Kaggle repository). The Adult dataset has 48842 samples and 14 features (6 numerical and 8 categorical). The Bank dataset has 45211 samples and 16 attributes (7 numerical and 9 categorical), and finally, the Kick dataset is the largest, with 72983 samples and 32 attributes, 14 numerical and 18 categorical. Table 4 presents the accuracy averaged over 5 random seeds and the standard deviation for deep learning models mentioned above, also including catBoost and XGBoost as baselines. Clearly, for binary classification tasks, deep learning models do not outperform gradient boosting methods such as CatBoost and XGBoost. It can also be observed that the proposed model achieves slightly lower accuracy than deep learning models, specifically 2% or 3% less depending on the model. However, Table 5 shows the training times of two of the deep learning models that achieve the highest accuracies, FT-Transformer and ResNet, compared to the neurovector-based model. It is worth noting the overhead introduced by FT-Transformer compared to the proposed model, reaching 8.5 times higher and even 10 times higher in the case of ResNet. For the Bank dataset, the training time of the neurovector-based neural network was 45 seconds, and for the larger Kick dataset, training was completed in 30 seconds. Considering the 2-3% difference in accuracy between the proposed neurovector model and the FT-Transformer or ResNet models, and the difference in training times in relation to $CO_2$ emissions produced, it can be concluded that the neurovector model is a more than suitable option for solving problems with tabular data.

## 5 CONCLUSIONS

In this work, a novel predictive algorithm for tabular datasets has been presented. The foundations of this algorithm are based on a new concept called neurovectors, which is a structured representation of data based on their tokenization. Thus, our approach redefines traditional machine learning paradigms by eliminating the need for hidden layers, backpropagation or weight adjustments. Due to the tokenization, a notable strength of the algorithm is its flexibility, it can effectively handle variables of any nature, from numerical to text, without any prior preprocessing and it is suitable for addressing both classification and prediction tasks, as has been shown. The experimental results using three well-known datasets have showed that neurovectors achieve competitive predictive

Table 4: Performance comparison for the proposed model and deep learning models

|  | Adult | Bank | Kick |
|---|---|---|---|
| SNN | 0.854 (0.0018) | 0.908 (0.0016) | 0.901 (0.0007) |
| Grownet | 0.857 (0.0019) | 0.909 (0.0012) | 0.902 (0.0006) |
| DCNv2 | 0.853 (0.0039) | 0.908 (0.0010) | 0.901 (0.0007) |
| AutoInt | 0.859 (0.0015) | 0.906 (0.0014) | 0.900 (0.0005) |
| MLP | 0.852 (0.0019) | 0.906 (0.0014) | 0.901 (0.0004) |
| ResNet | 0.854 (0.0017) | 0.907 (0.0014) | 0.902 (0.0005) |
| FT-Transformer | 0.859 (0.0010) | 0.909 (0.0014) | 0.902 (0.0003) |
| CatBoost | 0.873 (0.0012) | 0.907 (0.0015) | 0.902 (0.0009) |
| XGBoost | 0.872 (0.00046) | 0.909 (0.0009) | 0.903 (0.0003) |
| Neurovectors | 0.826 (0.3096) | 0.873 (0.2881) | 0.876 (0.1817) |

Table 5: Training times in seconds for Adult dataset

|  | Times | Overhead |
|---|---|---|
| ResNet | 144 | 9.6x |
| FT-Transformer | 128 | 8.5x |
| Neurovectors | 15 | 1x |

performance compared to conventional models such as random forest, gradient boosting, support vector machine and deep neural networks. In addition, the computational cost of the proposed neurovector-based learning architecture is on the order of $10^5$ FLOPs per prediction, two to four orders of magnitude lower than the FLOPs required by traditional machine learning models, which range from $10^7$ to $10^9$. Thus, the neurovectors-based predictive model offers a scalable, computationally efficient, and competitively accurate alternative to traditional machine learning and deep learning algorithms for tabular data.

Future work will focus on applying our approach to unstructured or semi-structured text data, a domain typically dominated by large language models. In fact, we investigate the behavior of the neurovectors algorithm in natural language processing tasks, exploring its potential to model language in a manner similar to large language models. Furthermore, it could be used not only for predictive tasks, which have been the focus of this work, but also to infer missing values or inverse prediction tasks.

## REPRODUCIBILITY STATEMENT

In order to ensure reproducibility, the authors have used public datasets and made the source code available in the GitHub repository at the following link: https://github.com/Neurovectores/module_anonimized_v2

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
