# OpenReview forum: "Learning Based on Neurovectors for Tabular Data: A New Neural Network Approach"
_ICLR.cc/2026/Conference — Submitted to ICLR 2026_

### Official Review · Reviewer_13uC · 2025-10-21

**Soundness:** 1
**Presentation:** 2
**Contribution:** 1
**Rating:** 2
**Confidence:** 4

**Summary:**

The paper present a predictive model based on Neurovectors, which are created to model similarity between data points. Unlike typical backpropagation used in neural network, the proposed model is trained usng energy-driven process. The model is evaluated on three tabular datasets and compared with basic baselines.

**Strengths:**

-The authors tackle an important problem of tabular data classification, where typical neural networks are comparable to shallow models.
-The idea is interesting and the model is inspired by LLMs

**Weaknesses:**

I could miss some important details but I think that the model is not correct. Looking at the formula of f(\tau) on page 4, there might not exist any neurovectors from the train set which have the same value at any feature. Take, for instance, a training set composed of two 2D points (1,1) and (2,2). If we want to make prediction for point (3,3), then what is returned by f(\tau)?

Even if the above could be corrected the method looks very similar to k-NN approach. Therefore I do not see much novelty in this method.

Finally, the evaluation is below the standards of ICLR: 3 simple datasets and only shallow (and one MLP) baselines is not sufficient. Even with strong novelty, the method has to be evaluated on more examples.

**Questions:**

The authors could explain points (1) from the weakness section. Moreover, they also should elaborate on the connections with k-NN.

---

> ### Author Response · Authors · 2025-11-18
> **answer to comment "the model is not correct..."**
>
> If we are in the training stage, that instance would become a neurovector, and if that occurs in the inference stage, any neurovector can be used, although a good prediction would not be expected in that case. However, this behavior is similar to the behavior of any prediction method that has to predict patterns that never occurred in the past.
>
> Even if the above could be corrected the method looks very similar to k-NN approach. Therefore I do not see much novelty in this method.

---

> ### Author Response · Authors · 2025-11-18
> **answer to comment "the method looks very similar to k-NN approach..."**
>
> We agree with the reviewers that there is a superficial analogy with k-NN in the inference phase and with an editing technique in the training phase. However, it has nothing to do with a knn enriched with an editing technique. First, we would like to emphasize that it is very likely that in most of the datasets used, applying a knn with the editing technique published by Wilson, D.L. (1972) is impossible due to its computational cost.
> Neurovector architecture introduces a dynamic learning mechanism that goes far beyond an editing technique. In the editing technique, the original dataset is edited and the weighting/elimination of instances is done statically. However, in our algorithm, the set of neurovectors is created dynamically only for instances that the model fails to predict (partial training), and the energy is updated dynamically with each correct prediction. This energy update means that the learning process requires several epochs  to obtain the final set of neurovectors. The concept we have called Energy is a key difference.  It is not just a question of which instances are retained (as in data editing), but how the historical reliability of each Neurovector is weighted. A Neurovector with a high count(NV) but low Energy will be less preferred than one with the same token match but higher Energy, allowing the model to learn which patterns are more reliable over time. Thus, the use of Energy as a confidence and tiebreaker metric, together with partial training (only on failures), establishes a learning mechanism that is fundamentally different from simple data editing or instance weighting in k-NN.
> On the other hand, the transformation of tabular data into a representation based on text string tokens and neurovectors is a very significant innovation, as it allows for a “text-like data” approach to learning, resulting in a highly efficient algorithm. Neurovectors are structures of interconnected nodes, and the search for candidate neurovectors is performed through tokens (text strings) indexed in a Python dictionary, resulting in an expected time of $O(1)$ per token. In terms of computing, our algorithm does not require dedicated resources (GPU), as it runs on a modern CPU in a standard computer
> We agree with the reviewer that our experimentation is not excessively broad, but our goal is not to present an algorithm that obtains the lowest error among the recent deep tabular baselines (FT-Transformer, TabPFN), since those algorithms are very computationally expensive (all these algorithms need GPU) and also not explainable.
>
> We present an algorithm with a much lower computational cost, which is explainable when tokenized and unifies machine learning theory since it can be used for any type of data and any type of task (whether classification or regression). For this reason alone, we believe that its presentation at ICLR would be very appropriate due to the novelty of the algorithm.

---

> ### Author Response · Authors · 2025-11-18
> **answer to comment "3 simple datasets and only shallow (and one MLP) baselines is not sufficient..."**
>
> We agree with the reviewer that our experimentation is not excessively broad, but our goal is not to present an algorithm that obtains the lowest error among the recent deep tabular baselines (FT-Transformer, TabPFN), since those algorithms are very computationally expensive (all these algorithms need GPU) and also not explainable.
>
> We present an algorithm with a much lower computational cost, which is explainable when tokenized and unifies machine learning theory since it can be used for any type of data and any type of task (whether classification or regression). For this reason alone, we believe that its presentation at ICLR would be very appropriate due to the novelty of the algorithm.

---

> ### Author Response · Authors · 2025-11-20
> **A revision has been uploaded**
>
> In this revision, the experimental evaluation has been extended using three new datasets and comparing with the algorithms SNN, Grownet, DCNv2, AutoInt, MLP, ResNet, F-Transformer, CatBoost and XGBoost. Two new tables (Tables 4 and 5) have been added. Table 4 compare the performance of the model proposed with recent deep learning models and Table 5 presents computing times. The following text has been added:
>
> "In order to compare the results of the neurovector-based algorithm with recent deep learning models designed specifically for tabular data, the results obtained by algorithms such as SNN, Grownet, DCNv2, AutoInt, MLP, ResNet and F-Transformer published in Gorishniy et al. (2021a) were considered using large datasets such as Adult Kohavi (1996), Bank Moro et al. (2014), and Kick (from Kaggle repository). The Adult dataset has 48842 samples and 14 features (6 numerical and 8 categorical).
> The Bank dataset has 45211 samples and 16 attributes (7 numerical and 9 categorical), and finally, the Kick dataset is the largest, with 72983 samples and 32 attributes, 14 numerical and 18 categorical. Table 4 presents the accuracy averaged over 5 random seeds and the standard deviation for deep learning models mentioned above, also including catBoost and XGBoost as baselines.
> Clearly, for binary classification tasks, deep learning models do not outperform gradient boosting methods such as CatBoost and XGBoost. It can also be observed that the proposed model achieves slightly lower accuracy than deep learning models, specifically 2% or 3% less depending on the model. However, Table 5 shows the training times of two of the deep learning models that achieve the highest accuracies, FT-Transformer and ResNet, compared to the neurovector-based model. It is worth noting the overhead introduced by FT-Transformer compared to the proposed model, reaching 8.5 times higher and even 10 times higher in the case of ResNet. For the Bank dataset, the training time of the neurovector-based neural network was 45 seconds, and for the larger Kick dataset, training was completed in 30 seconds. Considering the 2-3% difference in accuracy between the proposed neurovector model and the FT-Transformer or ResNet models, and the difference in training times in relation to CO2 emissions produced, it can be concluded that the neurovector model is a more than suitable option for solving problems with tabular data."

---

> > ### Comment · Reviewer_13uC · 2025-11-28
> > **Thanks for the answers**
> >
> > I appreciate the answers. I hope the authors conduct more extensive evaluation and improve the presentation of the methods to avoid misunderstanding. In the current form, the paper is not ready to be published in ICLR.

---

### Official Review · Reviewer_UCPG · 2025-10-27

**Soundness:** 2
**Presentation:** 2
**Contribution:** 1
**Rating:** 0
**Confidence:** 4

**Summary:**

This paper presents a novel supervised learning method and reports its accuracy on three small datasets. Unfortunately, the paper fails to point out that the new method is a variation of k-nearest neighbor with k=1.

**Strengths:**

The central idea is interesting and the experimental results are believable.

**Weaknesses:**

The major weakness is that the method proposed in this paper is not novel; it is a variation of k-nearest neighbor. Specifically, Equations 6 and 7 say that the predicted label of a test example is the label of the training example with maximum count(NV) score. The score of a training example is the number of its feature values that equal the value of the same feature in the test example.

The predicted label is the label of the single nearest (most similar) neighbor of the training example, where similarity is measured as the number of identical feature values.

Section 3.3 provides a method for editing the training set by upweighting examples that provide correct predictions. A conceptually similar idea is proposed by Wilson, D.L. (1972) Asymptotic properties of nearest neighbor rules using edited data. IEEE Transactions on Systems, Man, and Cybernetics, 2(3), 408-421.

Other weaknesses:

Equation 1 says that features are real-valued but then Section 3.2.1 requires exact matches, which is not sensible for real numbers.

Equations 8 and 9 are purely heuristic, so it is not justified to call the method energy-based.

The experiments are insufficient: on only three small datasets. The results in Table 2 are not impressive: the new method does not yield systematically better accuracy.

The paper claims that Python dictionary lookups have time complexity. This is true only in the average case, and the worst case is O(n).

The FLOPS discussion on page 8 is pointless because the datasets are all small. 10^10 FLOPS is less than a second on a GPU nowadays.

**Questions:**

No specific questions.

---

> ### Author Response · Authors · 2025-11-18
> **The algorithm based on neurovector is not a variation of KNN with a editing technique**
>
> We agree with the reviewers that there is a superficial analogy with k-NN in the inference phase and with an editing technique in the training phase. However, it has nothing to do with a knn enriched with an editing technique. First, we would like to emphasize that it is very likely that in most of the datasets used, applying a knn with the editing technique published by Wilson, D.L. (1972) is impossible due to its computational cost.
> Neurovector architecture introduces a dynamic learning mechanism that goes far beyond an editing technique. In the editing technique, the original dataset is edited and the weighting/elimination of instances is done statically. However, in our algorithm, the set of neurovectors is created dynamically only for instances that the model fails to predict (partial training), and the energy is updated dynamically with each correct prediction. This energy update means that the learning process requires several epochs  to obtain the final set of neurovectors. The concept we have called Energy is a key difference.  It is not just a question of which instances are retained (as in data editing), but how the historical reliability of each Neurovector is weighted. A Neurovector with a high count(NV) but low Energy will be less preferred than one with the same token match but higher Energy, allowing the model to learn which patterns are more reliable over time. Thus, the use of Energy as a confidence and tiebreaker metric, together with partial training (only on failures), establishes a learning mechanism that is fundamentally different from simple data editing or instance weighting in k-NN.
> On the other hand, the transformation of tabular data into a representation based on text string tokens and neurovectors is a very significant innovation, as it allows for a text-like data approach to learning, resulting in a highly efficient algorithm. Neurovectors are structures of interconnected nodes, and the search for candidate neurovectors is performed through tokens (text strings) indexed in a Python dictionary, resulting in an expected time of O(1) per token. In terms of computing, our algorithm does not require dedicated resources (GPU), as it runs on a modern CPU in a standard computer

---

> ### Author Response · Authors · 2025-11-18
> **how our algorithm works with real-valued features**
>
> The proposed learning algorithm describes a new neural network approach for tabular data, which addresses the handling of continuous (numerical) values in a fundamentally different way from traditional methods and recent deep learning methods with which the reviewer proposes we compare ourselves. The key to our algorithm's approach lies in tokenization and the representation of data as text, which eliminates the need for explicit preprocessing for both discrete and continuous variables. The Neurovectors method treats continuous values like any other type of data (categorical or text) through a process of transformation into text string tokens, and therefore no discretization is necessary. This is one of the main strengths and novelties, among others, of the method presented: the algorithm is designed to handle variables of any nature, from numeric to text, without any prior preprocessing.

---

> ### Author Response · Authors · 2025-11-18
> **answer to comment "Equations 8 and 9 are purely heuristic, so it is not justified to call the method energy-based"**
>
> We have called energy but other names could be also used such as reliability or trust.

---

> ### Author Response · Authors · 2025-11-18
> **answer to comment "The experiments are insufficient:..."**
>
> We agree with the reviewer that our experimentation is not excessively broad. Although we are expanding the comparison to address the reviewer's suggestions, our goal is not to present an algorithm that obtains the lowest error among the recent deep tabular baselines (FT-Transformer, TabPFN), since those algorithms are very computationally expensive (all these algorithms need GPU) and also not explainable.
>
>  We present an algorithm with a much lower computational cost, which is explainable when tokenized and unifies machine learning theory since it can be used for any type of data and any type of task (whether classification or regression). For this reason alone, we believe that its presentation at ICLR would be very appropriate due to the novelty of the algorithm.

---

> ### Author Response · Authors · 2025-11-18
> **answer to comments "The paper claims ..." and "The FLOPS discussion on..."**
>
> Our algorithm uses a text-like data approach to learning, resulting in a highly efficient algorithm. Neurovectors are structures of interconnected nodes, and the search for candidate neurovectors is performed through tokens (text strings) indexed in a Python dictionary, resulting in an expected time of $O(1)$ per token. In terms of computing, our algorithm does not require dedicated resources (GPU), as it runs on a modern CPU in a standard computer.

---

> ### Author Response · Authors · 2025-11-20
> **A revision has been uploaded**
>
> In this revision, the experimental evaluation has been extended using three new datasets and comparing with the algorithms SNN, Grownet, DCNv2, AutoInt, MLP, ResNet, F-Transformer, CatBoost and XGBoost. Two new tables (Tables 4 and 5) have been added. Table 4 compare the performance of the model proposed with recent deep learning models and Table 5 presents computing times. The following text has been added:
>
> "In order to compare the results of the neurovector-based algorithm with recent deep learning models designed specifically for tabular data, the results obtained by algorithms such as SNN, Grownet, DCNv2, AutoInt, MLP, ResNet and F-Transformer published in Gorishniy et al. (2021a) were considered using large datasets such as Adult Kohavi (1996), Bank Moro et al. (2014), and Kick (from Kaggle repository). The Adult dataset has 48842 samples and 14 features (6 numerical and 8 categorical).
> The Bank dataset has 45211 samples and 16 attributes (7 numerical and 9 categorical), and finally, the Kick dataset is the largest, with 72983 samples and 32 attributes, 14 numerical and 18 categorical. Table 4 presents the accuracy averaged over 5 random seeds and the standard deviation for deep learning models mentioned above, also including catBoost and XGBoost as baselines.
> Clearly, for binary classification tasks, deep learning models do not outperform gradient boosting methods such as CatBoost and XGBoost. It can also be observed that the proposed model achieves slightly lower accuracy than deep learning models, specifically 2% or 3% less depending on the model. However, Table 5 shows the training times of two of the deep learning models that achieve the highest accuracies, FT-Transformer and ResNet, compared to the neurovector-based model. It is worth noting the overhead introduced by FT-Transformer compared to the proposed model, reaching 8.5 times higher and even 10 times higher in the case of ResNet. For the Bank dataset, the training time of the neurovector-based neural network was 45 seconds, and for the larger Kick dataset, training was completed in 30 seconds. Considering the 2-3% difference in accuracy between the proposed neurovector model and the FT-Transformer or ResNet models, and the difference in training times in relation to CO2 emissions produced, it can be concluded that the neurovector model is a more than suitable option for solving problems with tabular data."

---

### Official Review · Reviewer_HGG6 · 2025-10-31

**Soundness:** 2
**Presentation:** 2
**Contribution:** 2
**Rating:** 2
**Confidence:** 4

**Summary:**

The paper introduces a **“neurovector”** paradigm for tabular learning that avoids backpropagation and trainable weights. Each training row is stored as a neurovector made of tokens from feature–value pairs; at inference, a test instance is tokenized the same way, candidate neurovectors are retrieved by token overlap, and the prediction is taken from the **most-overlapping** candidate. Formally, for test instance (x_j) with tokens (\tau_{j,l}), let (C_j) be the set of training neurovectors that share at least one token with (x_j). Let (M(NV,j)) be the number of shared tokens between a candidate (NV \in C_j) and (x_j). The method predicts
$$
m=\arg\max_{NV\in C_j} M(NV,j), \qquad \hat{y}_j = y_m .
$$

Ties are broken by an **energy** score. For classification:
$$
E(NV)=\frac{(s(NV))^2}{u(NV)} ,
$$
where (s(NV)) is the number of past correct uses of (NV) and (u(NV)) is the total uses. For regression:
$$
E_{\mathrm{reg}}(NV)=\frac{(s(NV))^2}{u(NV)} ,\exp!\left(-\alpha,\mathrm{MAE}(NV)\right).
$$

The approach aims to be **interpretable** (explicit token overlaps) and **efficient** (no gradient steps; create-on-error storage). On several UCI/Kaggle datasets, the authors report competitive accuracy compared to standard ML/DL baselines with reduced computational cost.

**Strengths:**

* **Simplicity & interpretability:** Prediction follows transparent token overlaps; energies provide per-instance diagnostics.
* **Gradient-free training:** Create-on-error storage avoids backpropagation/hyperparameter sweeps, attractive for low-resource settings.
* **Clear, reproducible core:** Retrieval and tie-breaking rules are explicit; basic complexity can be reasoned about via hash lookups and candidate ordering.
* **Potential efficiency:** If storage/candidates remain small, inference could be fast and memory-light in practice.

**Weaknesses:**

* **Limited evaluation:** Only a few datasets; no multi-seed cross-validation; several strong tabular baselines are missing (CatBoost/LightGBM/XGBoost, TabPFN, FT-Transformer); statistical tests and average-rank analyses are absent.
* **Tokenization brittleness:** Using **exact numeric values** as tokens risks near-zero overlap; discretization/quantization schemes (or similarity metrics) are not explored.
* **Compute claims unclear:** FLOP comparisons are indirect; no **wall-clock**, **RAM footprint**, or scaling curves vs. dataset size/feature cardinality; unclear training vs. inference accounting.
* **Protocol clarity:** Split definitions and tuning budgets per model are not consistently documented; potential for selection bias.
* **Theory gap:** No bounds on error, storage growth, or retrieval accuracy; the energy function lacks principled justification.

**Questions:**

1. **Numerical features:** How do you handle continuous values? Please report results with **binning/quantization** (e.g., equal-width, quantile, learned discretizers) and analyze sensitivity.
2. **Evaluation protocol:** Which split strategy is final (ratios, seeds)? Please provide **mean±std over 10–30 random splits** per dataset and **significance tests**.
3. **Compute & memory:** Report **wall-clock**, **RAM** (dictionary size vs. (|\text{train}|)), and scaling of candidate set size (m) with (|\text{train}|) and feature cardinality. Add **Pareto plots** (accuracy vs. compute/memory).
4. **Baselines:** Include **CatBoost/LightGBM/XGBoost** and recent deep tabular baselines (**FT-Transformer, TabPFN**) on a **larger benchmark suite** (10–20 datasets) with **average ranks**.
5. **Ablations:** (a) exponent in (\text{success}^2/\text{use}); (b) remove energy or replace with a **learned** tie-breaker; (c) **store-all** vs. **store-on-error**; (d) robustness to **label noise** and missing values.
6. **Collision control:** How are rare categories/high-cardinality handled? Any hashing scheme or token pruning?

---

> ### Author Response · Authors · 2025-11-18
> **answer to question #1**
>
> The proposed learning algorithm describes a new neural network approach for tabular data, which addresses the handling of continuous (numerical) values in a fundamentally different way from traditional methods and recent deep learning methods with which the reviewer proposes we compare ourselves. The key to our algorithm's approach lies in tokenization and the representation of data as text, which eliminates the need for explicit preprocessing for both discrete and continuous variables. The Neurovectors method treats continuous values like any other type of data (categorical or text) through a process of transformation into text string tokens, and therefore no discretization is necessary. This is one of the main strengths and novelties, among others, of the method presented: the algorithm is designed to handle variables of any nature, from numeric to text, without any prior preprocessing.

---

> ### Author Response · Authors · 2025-11-18
> **answer to question #3**
>
> In terms of computing, our algorithm does not require dedicated resources, as it runs on a modern CPU in a standard computer. This is because the tokens are stored as keys in a Python dictionary, which allows for efficient searching in $O(1)$ time. In fact, our training times for Adult dataset are 15 seconds versus 72 seconds or 187 seconds of ResNet and FT-Transformer, respectively (Table 10 from https://arxiv.org/pdf/2106.11959). In terms of memory, it should be noted that our algorithm has an internal mechanism for compressing the dataset through training, since only neurovectors with mispredicted instances are generated. As an example, in the wine dataset, out of a total of 4898 instances, 1230 neurovectors are generated, representing a 75% compression of the dataset. This results in much lower memory usage.

---

> ### Author Response · Authors · 2025-11-18
> **answer to Question#4**
>
> The XGBoost algorithm is included in Table 2 as Gradient Boosting.
>
> On the other hand, although we are expanding the comparison to address the reviewer's suggestions, our goal is not to present an algorithm that obtains the lowest error among the recent deep tabular baselines (FT-Transformer, TabPFN), since those algorithms are very computationally expensive and also not explainable.
>
>  We present an algorithm with a much lower computational cost, which is explainable when tokenized and unifies machine learning theory since it can be used for any type of data and any type of task (whether classification or regression). For this reason alone, we believe that its presentation at ICLR would be very appropriate due to the novelty of the algorithm.

---

> ### Author Response · Authors · 2025-11-18
> **answer to Question#6**
>
> The Neurovectors method does not require any special or explicit treatment for rare and high-cardinality categories, meaning that it is not necessary to apply dimensionality reduction techniques such as hashing or pruning. Its treatment is entirely implicit, mainly due to two aspects: first, the type of knowledge representation that our algorithm performs, including all unique values as text string tokens; and second, due to partial training in which neurovectors are generated with incorrectly predicted instances. Thus, in the case of a high-cardinality feature, it is very likely that this instance will result in a correct prediction and will not generate any new neurovectors. In the case of a rare feature, the partial training of the algorithm will cause instances containing rare features to result in incorrect predictions and therefore trigger the creation of a new neurovector for that instance. In this way, the redundancy of high cardinality is eliminated and the information of very low cardinality features is taken into account.

---

> ### Author Response · Authors · 2025-11-20
> **A revision has been uploaded**
>
> In this revision, the experimental evaluation has been extended using three new datasets and comparing with the algorithms SNN, Grownet, DCNv2, AutoInt, MLP, ResNet, F-Transformer, CatBoost and XGBoost. Two new tables (Tables 4 and 5) have been added. Table 4 compare the performance of the model proposed with recent deep learning models and Table 5 presents computing times. The following text has been added:
>
> "In order to compare the results of the neurovector-based algorithm with recent deep learning models designed specifically for tabular data, the results obtained by algorithms such as SNN, Grownet, DCNv2, AutoInt, MLP, ResNet and F-Transformer published in Gorishniy et al. (2021a) were considered using large datasets such as Adult Kohavi (1996), Bank Moro et al. (2014), and Kick (from Kaggle repository). The Adult dataset has 48842 samples and 14 features (6 numerical and 8 categorical).
> The Bank dataset has 45211 samples and 16 attributes (7 numerical and 9 categorical), and finally, the Kick dataset is the largest, with 72983 samples and 32 attributes, 14 numerical and 18 categorical. Table 4 presents the accuracy averaged over 5 random seeds and the standard deviation for deep learning models mentioned above, also including catBoost and XGBoost as baselines.
> Clearly, for binary classification tasks, deep learning models do not outperform gradient boosting methods such as CatBoost and XGBoost. It can also be observed that the proposed model achieves slightly lower accuracy than deep learning models, specifically 2% or 3% less depending on the model. However, Table 5 shows the training times of two of the deep learning models that achieve the highest accuracies, FT-Transformer and ResNet, compared to the neurovector-based model. It is worth noting the overhead introduced by FT-Transformer compared to the proposed model, reaching 8.5 times higher and even 10 times higher in the case of ResNet. For the Bank dataset, the training time of the neurovector-based neural network was 45 seconds, and for the larger Kick dataset, training was completed in 30 seconds. Considering the 2-3% difference in accuracy between the proposed neurovector model and the FT-Transformer or ResNet models, and the difference in training times in relation to CO2 emissions produced, it can be concluded that the neurovector model is a more than suitable option for solving problems with tabular data."

---

### Official Review · Reviewer_Mhyi · 2025-11-01

**Soundness:** 2
**Presentation:** 3
**Contribution:** 2
**Rating:** 2
**Confidence:** 3

**Summary:**

The paper proposes a novel learning paradigm for tabular data that replaces backpropagation with energy propagation in vector spaces. Instead of weight updates, the model encodes information through interconnected nodes and vector relationships, aiming for higher interpretability and computational efficiency.  Experimental results demonstrate the effectiveness of this method.

**Strengths:**

1. The paper presents a well-motivated idea. Transforming tabular data into vectorized or text-like representations to make them compatible with large language models (LLMs). This direction is timely and meaningful, as it moves beyond conventional tree-based models toward architectures that can leverage foundation models.
2. The paper is clearly written and well-structured, making the proposed approach easy to follow and conceptually accessible.

**Weaknesses:**

1. The experimental evaluation is rather limited in scope. The paper includes only a few datasets, and the results in Table 2 are not convincing. For instance, on Breast Cancer, the proposed method performs comparably to the baseline; on Absenteeism at Work, results are reported as N/A; and Red Wine Quality is a small, non-representative dataset. To substantiate the claimed advantages, additional experiments on more diverse and large-scale tabular datasets are necessary. Moreover, the paper omits comparisons with strong deep learning baselines specifically designed for tabular data, such as FT-Transformer, TabNet, or NODE. Including these methods would provide a more meaningful and fair evaluation of the proposed approach, particularly in assessing its scalability and competitiveness against modern deep architectures.
2. In the experiments, does “Gradient Boosting” refer to XGBoost, Gradient Boosting Decision Trees (GBDT), Gradient Boosting Regression Trees (GBRT), or another variant of the Gradient Boosting family? Please clarify which specific implementation or library was used, as different versions can differ substantially in optimization strategy, regularization, and performance.

**Questions:**

see above

---

> ### Author Response · Authors · 2025-11-18
> **answer to weaknesses #1**
>
> Although we are expanding the comparison to address the reviewer's suggestions, our goal is not to present an algorithm that obtains the lowest error among the recent deep tabular baselines (FT-Transformer, TabPFN or NODE), since those algorithms are very computationally expensive and also not explainable.
>
> In terms of computing, our algorithm does not require dedicated resources, as it runs on a modern CPU in a standard computer. This is because the tokens are stored as keys in a Python dictionary, which allows for efficient searching in $O(1)$ time. In fact, our training times for Adult dataset are 15 seconds versus 72 seconds or 187 seconds of ResNet and FT-Transformer, respectively (Table 10 from https://arxiv.org/pdf/2106.11959).
>
> On the other hand, our algorithm is not a black-box because to explain a prediction, it simply identifies the Neurovector that was selected and lists the exact features (tokens) that matched the test instance. This provides a clear and straightforward local explanation.
>
> Thus, we present an algorithm with a much lower computational cost, which is explainable when tokenized. In addition, our algorithm unifies machine learning theory since it can be used for any type of data and any type of task (whether classification or regression). For this reason alone, we believe that its presentation at ICLR would be very appropriate due to the novelty of the algorithm.

---

> ### Author Response · Authors · 2025-11-18
> **answer to weaknesses #2**
>
> The XGBoost algorithm is included in Table 2 as Gradient Boosting

---

> ### Author Response · Authors · 2025-11-20
> **A revision has been uploaded**
>
> In this revision, the experimental evaluation has been extended using three new datasets and comparing with the algorithms SNN, Grownet, DCNv2, AutoInt, MLP, ResNet, F-Transformer, CatBoost and XGBoost. Two new tables (Tables 4 and 5) have been added. Table 4 compare the performance of the model proposed with recent deep learning models and Table 5 presents computing times. The following text has been added:
>
> "In order to compare the results of the neurovector-based algorithm with recent deep learning models designed specifically for tabular data, the results obtained by algorithms such as SNN, Grownet, DCNv2, AutoInt, MLP, ResNet and F-Transformer published in Gorishniy et al. (2021a) were considered using large datasets such as Adult Kohavi (1996), Bank Moro et al. (2014), and Kick (from Kaggle repository). The Adult dataset has 48842 samples and 14 features (6 numerical and 8 categorical).
> The Bank dataset has 45211 samples and 16 attributes (7 numerical and 9 categorical), and finally, the Kick dataset is the largest, with 72983 samples and 32 attributes, 14 numerical and 18 categorical. Table 4 presents the accuracy averaged over 5 random seeds and the standard deviation for deep learning models mentioned above, also including catBoost and XGBoost as baselines.
> Clearly, for binary classification tasks, deep learning models do not outperform gradient boosting methods such as CatBoost and XGBoost. It can also be observed that the proposed model achieves slightly lower accuracy than deep learning models, specifically 2% or 3% less depending on the model. However, Table 5 shows the training times of two of the deep learning models that achieve the highest accuracies, FT-Transformer and ResNet, compared to the neurovector-based model. It is worth noting the overhead introduced by FT-Transformer compared to the proposed model, reaching 8.5 times higher and even 10 times higher in the case of ResNet. For the Bank dataset, the training time of the neurovector-based neural network was 45 seconds, and for the larger Kick dataset, training was completed in 30 seconds. Considering the 2-3% difference in accuracy between the proposed neurovector model and the FT-Transformer or ResNet models, and the difference in training times in relation to CO2 emissions produced, it can be concluded that the neurovector model is a more than suitable option for solving problems with tabular data."

---

### Meta-Review · Area_Chair_vREk · 2025-12-10

**Summary:**

The authors propose a new learning approach for tabular data based on neurovectors. The reviewers acknowledge the well-motivated idea, its simplicity, and the clear writing and structure. However, the reviewers had crucial concerns, mainly the limited novelty (similarity with k-NN), errors in the method formulation, and poor benchmarking along with weak baselines.

**Reviewer Concerns:**

The authors did not post a rebuttal, so all concerns remain outstanding.

**Reviewer Scores:**

Since there was no rebuttal posted by the authors, the scores would either remain the same or even decrease.

---

### Decision · Program_Chairs · 2026-01-26

Reject